# Estimating 500-m Resolution Soil Moisture Using Sentinel-1 and Optical Data Synergy

**Myriam Foucras [1], Mehrez Zribi [1,\*] , Clément Albergel [2] , Nicolas Baghdadi [3] , Jean-Christophe Calvet [2] and Thierry Pellarin [4]**

[1] CESBIO, Université de Toulouse, CNRS/UPS/IRD/CNES/INRAE, 18 Avenue Edouard Belin, bpi 2801, 31401 Toulouse CEDEX 9, France; foucrasm@cesbio.cnes.fr

[2] CNRM, Université de Toulouse, Meteo-France, CNRS, 31057 Toulouse, France; clement.albergel@meteo.fr (C.A.); jean-christophe.calvet@meteo.fr (J.-C.C.)

[3] INRAE, TETIS, University of Montpellier, 500 rue François Breton, 34093 Montpellier CEDEX 5, France; nicolas.baghdadi@teledetection.fr

[4] CNRS, IRD, University Grenoble Alpes, Grenoble INP, IGE, F-38000 Grenoble, France; thierry.pellarin@univ-grenoble-alpes.fr

\* Correspondence: mehrez.zribi@ird.fr

**Abstract:** The aim of this study is to estimate surface soil moisture at a spatial resolution of 500 m and a temporal resolution of at least 6 days, by combining remote sensing data from Sentinel-1 and optical data from Sentinel-2 and MODIS (Moderate-Resolution Imaging Spectroradiometer). The proposed methodology is based on the change detection technique, applied to a series of measurements over a three-year period (2015 to 2018). The algorithm described here as "Soil Moisture Estimations from the Synergy of Sentinel-1 and optical sensors (SMES)" proposes different options, allowing information from vegetation densities and seasonal conditions to be taken into account. The output from this algorithm is a moisture index ranging between 0 and 1, with 0 corresponding to the driest soils and 1 to the wettest soils. This methodology has been tested at different test sites (South of France, Central Tunisia, Western Benin and Southwestern Niger), characterized by a wide range of different climatic conditions. The resulting surface soil moisture estimations are compared with in situ measurements and already existing satellite-derived soil moisture ASCAT (Advanced SCATterometer) products. They are found to be well correlated, for the African regions in particular (RMSE below 6 vol.%). This outcome indicates that the proposed algorithm can be used with confidence to estimate the surface soil moisture of a wide range of climatically different sites.

**Keywords:** change detection algorithm; Sentinel-1; soil moisture; MODIS; ASCAT; Sentinel-2

## 1. Introduction

Surface soil moisture plays a key role in the continental water cycle, and is a fundamental parameter characterizing the manner in which rainwater is shared between the three major processes: runoff, infiltration and evapotranspiration [1–6]. It is also a very important parameter for irrigation management and the optimization of crop needs [7]. When the surface soil moisture content needs to be evaluated, in situ measurements are often carried out over a small selection of agricultural fields. However, these measurements are expensive and do not match the requirements of fine spatial sampling, especially in contexts where the soil water content can have a high spatial variability due to heterogeneities in precipitation, irrigation, soil texture, etc.

It is in this context that satellite remote sensing has for several years shown its potential for the estimation of this parameter [8–10]. Low spatial resolution measurements are the most advanced in

operational terms. Indeed, there are several products that can be used to access global maps of surface soil moisture, at a spatial resolution greater than 10 km and a temporal resolution close to 1 day [11–18]. These products are based mainly on passive microwave or radar scatterometer observations. In the case of the former, the SMOS (Soil Moisture and Ocean Salinity) [15,16] and SMAP (Soil Moisture Active Passive) [16,17] space missions are significant since they were designed and developed specifically for the measurement of surface soil moisture.

In the last 30 years, SAR (Synthetic Aperture Radar) remote sensing has demonstrated its considerable potential for the estimation of soil moisture at high spatial resolutions [18–28]. Numerous studies have improved our understanding of the behavior of multi-configuration radar signals, through the experimental analysis of radar data acquired over land surfaces [29,30] as well as the development of various backscattering models. The latter are relevant to the study of bare soils, and can be classed as physical [31], semi-empirical, or empirical models [32–34]. They are sensitive to the surface soil moisture, which is directly related to the dielectric constant. In the case of soils covered by vegetation, there are also many physical and semi-empirical models. The most commonly used of these is the WCM (Water–Cloud Model) [35,36]. Soil moisture can be retrieved by inverting the radar data, using various different approaches. These tend to vary according to the characteristics of each specific SAR mission (ERS, ALOS, ENVISAT, RADARSAT, TERRASAR-X, etc.). The simplest model makes use of a single radar configuration [37], whereas more complex models take multi-frequency [38–40], multi-angular [41,42], or polarimetric data [43–46] into account.

With the launch of the Sentinel-1 constellation, the opportunity has arisen for the development of operational algorithms, and several very high spatial resolution products have been proposed to meet agricultural needs at the scale of individual plots. However, this improvement in spatial resolution has been achieved at the expense of lower temporal resolutions (typically corresponding to a 6-day repeat cycle) [47–50]. The two main approaches used to analyze the raw data are known as the neural network and change detection techniques. El-Hajj et al. [47] proposed an algorithm based on neural networks, using coupling between the IEM (Integral Equation Model) and WCM models to train simulated radar signals over soil surfaces with vegetation cover. Gao et al. [49] proposed a change detection algorithm that makes use of the influence of surface soil moisture on radar signal dynamics, and which is also sensitive to the density of the canopy. Despite the considerable demand for surface soil moisture products at the scale of individual plots, these techniques are very demanding in terms of computing time and memory, thus making them unduly complex and unsuitable for the large-scale mapping of surface soil moisture.

On the other hand, an intermediate resolution close to 1 km appears to be suitable for the analysis of hydrological processes in a regional context. This scale has thus been addressed in various studies [51–54]. This intermediate spatial scale can be used to bridge the gap between high-resolution and low-resolution products. The first ones need important information about the land use and vegetation properties at the field scale to retrieve the soil effects on a radar signal, with an important capacity of computing for large regions. Low resolution offers very useful information at very large scales (continental, global, etc.), but is not accurate enough to analyze hydrological processes at the regional scale.

Several studies, based on the disaggregation of the surface soil moisture condition at a low resolution, have estimated surface soil moisture at high resolutions, using thermal infrared data such as that provided by MODIS (MODerate resolution Imaging Spectroradiometer). Merlin et al. [51] proposed the DISPATCH (DISaggregation based on Physical And Theoretical scale CHange) algorithm, based on the use of SMOS measurements and MODIS data, which has been validated for different climatic conditions and landscapes. More recent studies have proposed surface soil moisture applications based on the analysis of radar data produced by ESA's new Sentinel-1 constellation. Bauer-Marschallinger et al. [53] proposed a change detection algorithm, which allows surface soil moisture to be estimated from Sentinel-1 data at a scale of 1 km using 10 m spatial resolution data. Indeed, using data provided at a 10 m spatial resolution is very useful to correct and filter a 1 km

pixel data that may cover several land uses. This algorithm has been tested on different sites, and was found to provide accurate results over agricultural areas. Although this operational algorithm is very useful for the production of intermediate resolution surface soil moisture maps, its overall approach remains quite general and does not allow refinements or data corrections to be implemented for specific types of landscapes. In addition, some limitations have been observed in the case of heterogeneous surfaces [55]. In particular, over areas containing forests and vineyards, the Bauer-Marschallinger algorithm tends to overestimate the surface soil moisture. In addition, in the case of well-developed vegetation cover, it produces underestimated SSM (Surface Soil Moisture) values [55].

In this context, the present paper describes an algorithm based on the change detection technique. The novelty of this method can be attributed to two of its characteristics: the relative sophistication of the proposed algorithm and the spatial resolution (500 m) of the soil moisture product. The present study provides a medium resolution (~1 km) product. The challenge with this approach is that of minimizing the SM errors induced by local heterogeneities. As explained in Section 3, the proposed algorithm includes various processes that have been designed to assess, and then compensate for, factors such as land cover, vegetation cover density, and the seasonal context of the surface soil moisture dynamics. Three different options are proposed, to enable the analysis of different geographical regions.

This paper is organized as follows: Section 2 is dedicated to the description of the satellite database and the validation sites; Section 3 describes the proposed methodology; Section 4 presents the results, together with our discussion of the surface soil moisture estimations derived for several study sites, which are compared to those provided by ASCAT; and our conclusions are presented in Section 5.

## 2. Study Areas and Dataset Description

### 2.1. Study Areas

In the present study, several areas were investigated for two reasons: firstly, it was important to understand the behavior of surface soil moisture as a function of the local climate and soil properties in contrasted areas; secondly, it was important to use in situ ground truth measurements to check the accuracy of the surface soil moisture estimations derived from satellite data. The studied regions, situated in the South of France (the Occitania region), West Africa, and North Africa, are shown in Figure 1. These areas were accessible and equipped with ground stations.

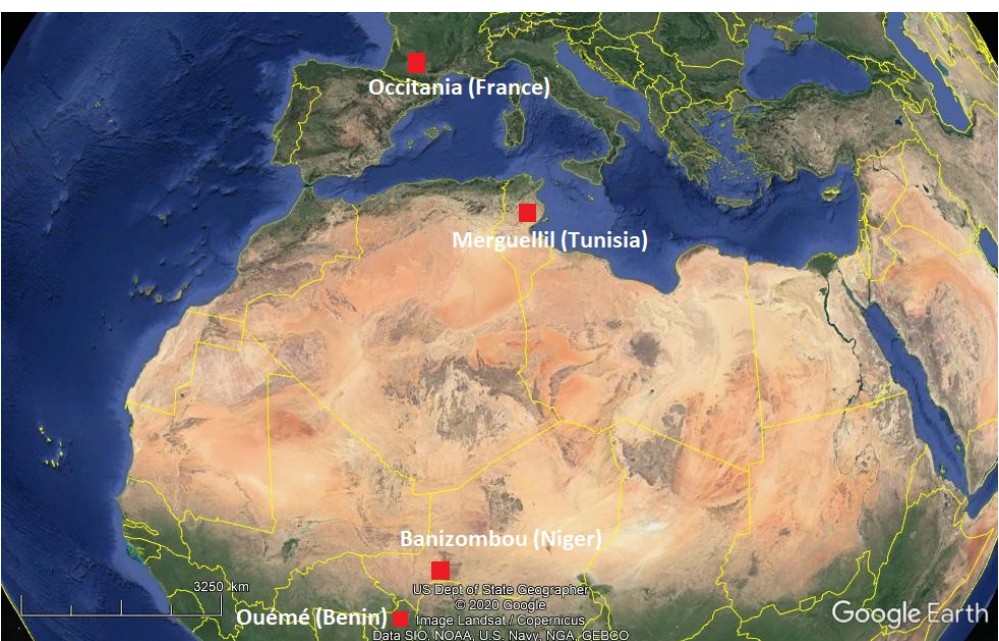

**Figure 1.** Studied areas (Occitania, Merguellil, Banizombou, and Ouémé).

### 2.1.1. The Occitania Region (South of France)

The Occitania region was studied at several different sites, close to the cities of Toulouse and Montpellier. The in situ surface soil moisture measurements were provided by the SMOSMANIA observing system. SMOSMANIA is a long-term campaign that has been organized in an effort to acquire surface soil moisture profiles from 21 automated weather stations in southwestern and southeastern France [56,57]. It began in 2007 and was integrated into the real-time Météo-France observation system in 2014. This system is based on Météo-France's existing automatic weather station network, and was developed for the validation and modeling of remotely sensed surface soil moisture estimations. The stations were chosen to form a Mediterranean–Atlantic transect, in order to study the marked climatic gradient between the two coastlines. The surface soil moisture probes (ThetaProbe of Delta T Device) were calibrated at all depths (5, 10, 20, and 30 cm) by measuring surface soil moisture from gravimetric soil samples collected during the installation. In this study, only measurements at a depth of 5 cm were used. While this region mainly consists of croplands, the stations are generally covered by grass. Three stations (at Savènes, Mouthoumet, and Narbonne), which are representative of the three different climates and landcover types in the Occitania region (with low forest densities only, since forested land cover is not well suited to the testing of our algorithm), were analyzed in this study. Their mean temperatures ranged between 12.3 and 15.2 °C, and their mean annual precipitation ranged between 649 and 845 mm. Data can be obtained from the International Soil Moisture Network (https://ismn.geo.tuwien.ac.at/en/).

### 2.1.2. Merguellil Site

This study site is situated in the Kairouan plain in Central Tunisia [22]. The climate in this region is semi-arid, with an average annual rainfall of approximately 300 mm/year, characterized by a rainy season lasting from October to May, with the two rainiest months being October and March. As is generally the case in semi-arid areas, the rainfall patterns in this area are highly variable over time and space. The mean temperature in Kairouan city is 19.2 °C (minimum of 10.7 °C in January and maximum of 28.6 °C in August). The mean annual potential evapotranspiration (Penman) is close to 1600 mm, and the landscape is mainly flat.

Over this study site, a network of continuous thetaprobe stations, installed in bare soil locations, provided moisture measurements at depths of 5 and 40 cm. All measurements were calibrated using gravimetric estimations [22]. Data were obtained from http://osr-cesbio.ups-tlse.fr/.

### 2.1.3. Niger

The studied region is located in the southwestern part of Niger, between the Niger river and the fossil valley of Dallo Bosso [58]. The Sahelian climate in this region is semi-arid, with an average annual rainfall ranging between 300 and 750 mm, characterized by a rainy season from June to September. The landscape is mainly flat and dominated by a dissected plateau with slopes of less than 6%, which have lateritic soils and are covered with tiger bush. These are bordered by terrain with mainly strong transitional features and steep inclines, which can reach a slope of 35%. The vegetation in the valleys is dominated by both fallow and cultivated fields (mainly millet). Continuous ground measurements were carried out at a depth of 5 cm at the Banizombou site. Data can be obtained from the International Soil Moisture Network (https://ismn.geo.tuwien.ac.at/en/).

### 2.1.4. Benin

The Ouémé catchment is located in Benin, West Africa, and is a part of the AMMA–CATCH observatory (African Monsoon Multidisciplinary Analysis—Coupling the Tropical Atmosphere and the Hydrological Cycle [58]; http://www.amma-catch.org), whose mission is to study the hydrological impact of climate and anthropogenic changes. With a surface area of 12,000 km$^2$, the Ouémé catchment is covered mainly by savanna, forests, and crops. The rainy season lasts from April to October, and its

annual rainfall is approximately 1250 mm. The Ouémé basin has a very dense network of instruments, which are used to monitor the water cycle and the vegetation dynamics in this sub-humid region. Surface soil moisture is measured continuously at a depth of 5 cm. Data can be obtained from the International Soil Moisture Network (https://ismn.geo.tuwien.ac.at/en/).

*2.2. Satellite Dataset*

2.2.1. Sentinel-1

Sentinel-1A (S-1A) and Sentinel-1B (S-1B) were launched in April 2014 and April 2016, respectively, as part of the European Space Agency's program "Copernicus", designed for the observation and monitoring of the Earth's surface and providing operational applications with environmental information. Synthetic Aperture Radars (SARs) are indifferent to weather conditions and allow data to be acquired at any time of the day or night. The SAR payloads on each of these satellites, which are positioned 180° apart in a sun-synchronous orbital plane, provide single and dual polarization images in the C-band (wavelength approximately 6 cm). Ascending mode images recorded from several (in general 2 or 3) successive orbits are used to produce "multi-looked" (i.e., viewed from several different incidence angles) Level-1 Ground Range Detected (GRD) products, with a spatial resolution of 10 m, every six days. These products were obtained from the Copernicus website (https://scihub.copernicus.eu/dhus/#/home). Several processing steps were performed on each image in order to extract the backscattering coefficient.

In the present study, images of the selected study sites, recorded in the VV and VH polarizations, were used. These included both descending orbit images taken in the morning (between 5 a.m. and 7 a.m.) and ascending orbit images taken 12 h later (between 5 p.m. and 7 p.m.). The number of usable Sentinel-1 images depends strongly on the chosen area: just 120 images are produced per year for Benin and Niger, as opposed to more than 400 images per year for the Tunisian and French sites. It is important to note that although the Sentinel-1 observations are insensitive to the local weather conditions, as a consequence of the orbital geometry and the resulting viewing angle from which a given site can be observed, some of the images are incomplete. This can considerably reduce the number of Sentinel-1 observations that can be used to provide data for a given pixel in the processed image. Furthermore, there are some areas of Africa for which no descending orbit images are available.

2.2.2. Sentinel-2

Sentinel-2 is a land-monitoring mission developed by the European Space Agency (ESA), which comprises a constellation of two multi-spectral imaging satellites, Sentinel-2A and Sentinel-2B. Sentinel-2A (S-2A) was launched in June 2015 and was followed by Sentinel-2B (S-2B) in March 2017. This constellation provides global coverage of the Earth's land surface with a revisit frequency of 5 days. The images are produced in 13 spectral bands, covering visible and mid-infrared wavelengths, at three different spatial resolutions (10, 20, and 60 m). The optical images used for the present study were obtained from the THEIA website at the French Land Data Center (https://www.theia-land.fr/), which makes the data available in the so-called "Level-2A" format. This product includes radiometric, atmospheric, and cloud corrections, based on an algorithm developed by [59]. The THEIA website does not provide Level-2A images of the African continent.

2.2.3. MODIS

Over the last 20 years, the two NASA satellites "Terra" and "Aqua", both of which carry the MODIS (MODerate resolution Imaging Spectroradiometer) instrument, have provided moderate resolution spectral images of the Earth. In the present study, a 250 m spatial resolution (the finest available from MODIS) and 16-day composites of NDVI data (MOD13Q1 and MYD13Q1) were used in order to optimize the likelihood of including the best quality pixels (minimal cloud cover). The MODIS products were provided by NASA (https://earthdata.nasa.gov/). This product is retrieved

from atmosphere-corrected, daily bidirectional surface reflectance observations, using a compositing technique based on product quality assurance metrics to remove low quality pixels. The map of Figure 1 shows the different study areas. By combining the data provided by both satellites, an annual total of 46 images of these tiles can be produced.

Sentinel-2 and MODIS both provide NDVI products, at different temporal and spatial scales. The Sentinel-2 pixels are 625 times smaller than those produced by MODIS. However, whereas the Sentinel-2 images can be unusable for several weeks at a time, for example during the cloudy conditions of the African monsoon, in general at least one or two good quality composite MODIS images can be produced per month.

### 2.2.4. ASCAT

The ASCAT radar is one of the instruments carried by ESA's three Metop satellites. ASCAT soil moisture products, recorded over a 25 km swath grid, can be recovered from the Earth Observation Portal (https://eoportal.eumetsat.int/). ASCAT is operated in the C-band (5.3 GHz) and in the vertical polarization. Over land, the measured radar backscattering coefficient depends on the surface soil moisture, surface roughness, vegetation characteristics and incidence angles of the transmitted radar beam. Surface soil moisture data is retrieved from the backscattering coefficient, using a change detection method [10]. In the framework of the soil moisture CCI (Climate Change Initiative), the European Space Agency has proposed surface soil moisture estimates based on the combined use of these different measurements. The ASCAT soil moisture considered product "ASCAT Soil Moisture at 25 km Swath Grid—Metop" is not based on a grid as Sentinel or MODIS. So, in our study, for each site and for each day with available data, an average value is computed over ASCAT measurements in a region of +/−0.05° in latitude and longitude around the site.

Table 1 summarizes the characteristics of the various types of satellite data addressed in the present study.

**Table 1.** Summary of satellite data characteristics.

|  | Sentinel 1 | Sentinel 2 | MODIS | ASCAT |
|---|---|---|---|---|
| Satellite launch | S-1A: 3 April 2014<br>S-1B: 25 April 2016 | S-2A: 23 June 2015<br>S-2B: 7 March 2017 | 18 December 1999 | 19 October 2006 |
| Image type | Radar<br>2 orbits (ASC/DES)<br>2 polarizations (VV/VH) | Optical | Optical | Radar |
| Temporal resolution | 6 or 12 days<br>Some gaps due to orbit geometry | 5 or 6 days<br>Some gaps due to weather conditions | 8 days<br>Some gaps due to weather conditions | 3 days |
| Spatial resolution | 10 m | 10 m | 250 m | 25 km |

Due to the availability of the satellite data, the study period covers a 3-year period, including 2016, 2017, and 2018. The data from 2019 was not analyzed because the corresponding ground measurements need correction processing, which can take several months, and thus were not available at the time of the study.

In addition, Table 2 provides the characteristics of each site of interest. For each site, the GPS coordinates (the French sites' positions are confidential) and associated Sentinel and MODIS tiles are listed.

**Table 2.** Characteristics of the sites of interest.

| Country | Closest City | ID | GPS Coordinates | Sentinel Tile | MODIS Tile |
|---------|--------------|-----|-----------------|---------------|------------|
| France | Savenes | SVN | - | 31TCJ | h18v04 |
| France | Narbonne | NBN | - | 31TDH | h18v04 |
| France | Mouthoumet | MTM | - | 31TDH | h18v04 |
| Tunisia | Hmidate | HMD | 35.48° N 9.84° E | 32SNE | h18v05 |
| Tunisia | INGC | INGC | 35.62° N 9.94° E | 32SNE | h18v05 |
| Niger | Niamey | BZ1 | 13.56° N 2.66° E | 31PDR | h18v07 |
| Benin | Djougou | BLM | 9.74° N 1.60° E | 31PCL | h18v08 |

## 3. Methodology

In this section, we present the methodology used to develop a radar data inversion algorithm to provide surface soil moisture estimations, with a spatial resolution of 500 m and a temporal resolution of 6 days. The proposed algorithm is called SMES: Soil Moisture Estimations from the Synergy of Sentinel-1 and optical sensors.

### 3.1. General Methodology

The aim of this study was to develop an algorithm to provide surface soil moisture estimations with a spatial resolution of 500 m. The first step involves the aggregation of Sentinel-1 pixels, in order to generate radar images with a spatial resolution of 500 m (by taking the average value of 10 m radar signal pixels, inside each 500 m pixel). For this, average values of surface soil moisture are computed, using the methodology described in [10]. A soil moisture index, ranging in value between 0 and 1 (with 0 corresponding to the driest soil conditions and 1 representing the wettest soil conditions) is computed as follows:

$$\mathcal{I}_t = \frac{\sigma_t - \sigma_{min}}{\sigma_{max} - \sigma_{min}} \tag{1}$$

where:

- $\mathcal{I}_t$ is the moisture index at time $t$;
- $\sigma_t$ is the radar signal in dB at time $t$ for a given 500 m × 500 m pixel;
- $\sigma_{min}$ and $\sigma_{max}$ are, respectively, the minimum and maximum radar signals in dB, for any given pixel.

In order to compare the moisture index $\mathcal{I}_t$ with ground measurements, moisture index products are converted to physical units of m$^3$ m$^{-3}$:

$$\theta_t = \theta_{min} + \mathcal{I}_t \times (\theta_{max} - \theta_{min}) \tag{2}$$

where:

- $\theta_t$ is the surface soil moisture content at time $t$ (m$^3$ m$^{-3}$);
- $\theta_{min}$ and $\theta_{max}$ represent the minimum and maximum values of in situ surface soil moisture at a depth of 5 cm at a given site (m$^3$ m$^{-3}$), defined by the 90% confidence interval of a Gaussian distribution [60]. By defining $\mu_\theta$ and $\sigma_\theta$ as the mean and standard deviation of the ground truth data over the 3-year period used for this analysis, $\theta_{min}$ and $\theta_{max}$ can be computed as follows: $\theta_{min} = \max(\mu_\theta - 1.65 \times \sigma_\theta, \min(\theta))$ and $\theta_{max} = \min(\mu_\theta + 1.65 \times \sigma_\theta, \max(\theta))$, where 1.65 represents the 95% quantile of the standard normal distribution. It was preferred to use these quantities rather than strict minimum and maximum values, in order to eliminate outliers. At the proposed spatial resolution, we consider the temporal influence of roughness on the radar signals to be constant.

　　This general methodology was used to compute the surface soil moisture over our study sites. However, various limitations were observed, when the same algorithm was applied over sites where the temporal dynamics of the water cycle and land cover were not the same. In an effort to adapt these surface soil moisture estimations to the local conditions, the following criteria affecting the properties of remotely sensed signals and soil characteristics should be considered:

- Radar signals are practically insensitive to surface soil moisture in the case of land cover such as forests and urban areas. For this reason, the radar signal data should be filtered and corrected, in order to consider only those areas that are suitable for surface soil moisture estimations.
- As radar signals are highly sensitive to the dynamics of the vegetation, a radar backscattering value should not be interpreted in the same way for different vegetation densities. For this reason, a more realistic analysis could be achieved with the change detection approach, using two different vegetation classes: one with a scattered vegetation cover and the other with a dense vegetation cover.
- The moisture cycle and the evaporative potential are not the same during the cold and hot seasons. This property is particularly visible in the moisture variations observed in the first centimeters of soil. In order to adjust the surface soil moisture estimations to seasonal conditions, the conversion of the moisture index to volumetric moisture could be adapted for each of these two seasons.

　　Each of these three processes is described in the following three sections.

### 3.2. Radar Signal Correction

　　Radar signal corrections can be applied to remove the influence of forests and urban areas, which are not useful for the surface soil moisture estimations.

　　The CESBIO (Centre d'Etudes Spatiales de la BIOsphère) produces land occupation maps for France at a resolution of 10 m, where each pixel is classified into one of 16 different classes (for example, water, deciduous forest, industrial area, etc.). A sample for the Occitania region is shown in Figure 2. From the initial 16 classes, 4 global classes were derived: urban zones, bodies of water, agricultural areas, and forests.

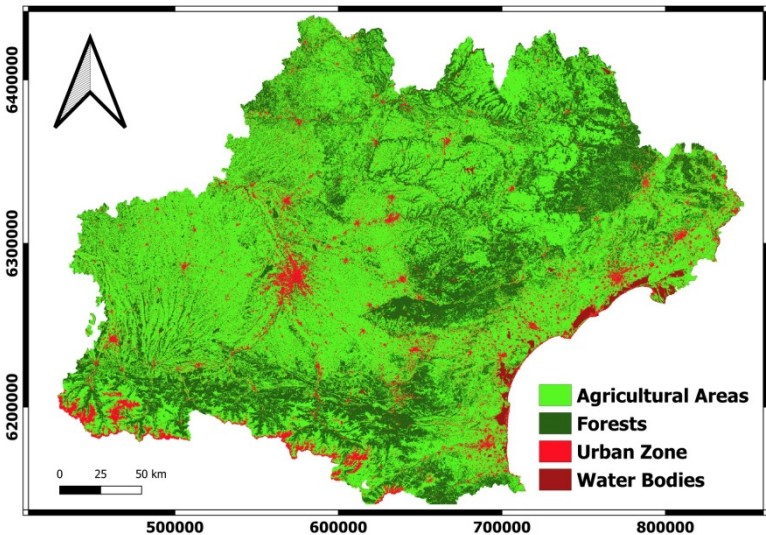

**Figure 2.** Land occupation map in the Occitania region.

　　The proportion of this area covered by bodies of water is very low and is near the Mediterranean Sea. The white pixels represent urban zones, and the city of Toulouse can be seen in the western portion of this figure. The density of urban zones is also higher close to the Mediterranean Sea. Close observation of this land cover map reveals a high degree of heterogeneity in the Occitania region,

which implies that a 500 m × 500 m pixel could in some cases include an urban zone, an agricultural area, and a portion of forest.

For this, four types of landscape were considered: urban areas, forest areas, areas of water, and finally the last one was composed of the agricultural landscape and natural areas. This classification is provided by the land cover map. It was decided to eliminate three types of landscapes:

- Forest areas: Sentinel-1 C band data over these areas is only slightly sensitive to the soil water content, due to the strong attenuation of radar signals by the vegetation cover. For this reason, failure to remove forest-covered areas could generate inaccuracies in the estimation of surface soil moisture.
- Water bodies: as these have a very weak radar signature, due to the specular reflections produced by their air–water interface, an offset of −20 dB is applied to the radar signals [53].
- Urban areas: these have radar signatures which vary only slightly throughout the year, and remain only weakly sensitive to soil water content, due to the relatively low percentage of bare soil in urban areas. For these reasons, it was considered inappropriate to include these areas when estimating the dynamics of surface soil moisture content.

In the following, we consider an example: the case of the Savenès station. In the 500 m × 500 m pixel centered on the Savenès station, 17% of the area is classified as forest and 5% as urban. It would be incorrect to take the average value of the radar signals over the total area, since only agricultural area should be considered. For this reason, the corrected radar signal should be computed by excluding urban and forest areas.

Over a 500 m × 500 m pixel, there are approximately 2100 Sentinel 1 pixels. The corrected radar signal is thus computed as the average value of the radar signals corresponding to the Sentinel-1 pixels that are not classed as forest, urban, or water:

$$\sigma_{\text{other}} \approx \frac{1}{n} \sum \sigma_{\text{no urban, no forest, no water}} \tag{3}$$

where:

- $\sigma_{\text{other}}$ is the average value of the radar signal over a 500 m × 500 m pixel, corresponding to mainly agricultural areas;
- $n$ is the number of Sentinel-1 pixels, which are not classed as urban, forest or water in the 500 m × 500 m pixel;
- $\sigma_{\text{no urban , no forest , no water}}$ is the radar signal over the pixels classed as agricultural areas.

For all of the French sites (see Table 2), the backscatter radar signal used for the SMES algorithm is computed using Equation (3). For the Nigerian, Tunisian, and Beninese sites, there is no need to consider the pixel classifications since there are only agricultural fields in these study areas. For this reason, the average value of the full 500 m × 500 m area can be applied.

### 3.3. Use of the Vegetation Density Information

As described in Section 3.1, the vegetation can have a significant influence on the backscatter radar signals. The approach proposed in this study associates the vegetation density information with changes in the radar signal. This can be done by distinguishing between two vegetation classes: one corresponding to bare soil or a low vegetation density, and the other corresponding to dense vegetation. We have used NDVI to build these two classes, with a low NDVI value corresponding to the first vegetation class, and a high value corresponding to the second class. The median value of the NDVI over the time and for each pixel was used to create these two vegetation classes, with an equal number of dates in each group. For this reason, a different value of "median NDVI" was associated with each 500 m × 500 m pixel, depending on the characteristics of the vegetation in each pixel. Following this initial step, each 500 m × 500 m pixel can be associated with a radar time series, classified into two

vegetation classes. This approach makes it straightforward to compute a pair of radar signal extrema ($\sigma_{\min}$ and $\sigma_{max}$) for low vegetation density areas and a second pair for high vegetation density areas. In Equation (1), the appropriate pair of radar signals ($\sigma_{min}$ and $\sigma_{max}$) is used.

In this case, the first step is to compute the NDVI value, taking into account only areas that are sensitive to surface soil moisture. As explained in the section on radar signal correction, it appears to be useful to remove forest and urban areas when computing the NDVI. In order to correct the NDVI computed from MODIS data, to produce a value referred to as $NDVI_{\text{other}}$ (corresponding essentially to the NDVI of agricultural areas), we use a combination of MODIS and Sentinel-2 products to achieve the best possible spatial and temporal resolutions for this parameter.

The NDVI derived from MODIS and Sentinel-2 data were first compared, as shown in Figure 3, confirming that they have very similar trends. In the case of the 31TCJ tile (in the vicinity of Toulouse), the MODIS NDVI was averaged over all pixels, in order to provide one value per day. Similarly, the Sentinel-2 NDVI was averaged, and then compared to the averaged MODIS NDVI values. It can be seen that these averaged NDVI values follow the same trend, with a RMSE of 0.08. The differences between these two sets of NDVI values can be accounted for by differences in data correction methodology and instrumental configuration (incidence angle, etc.).

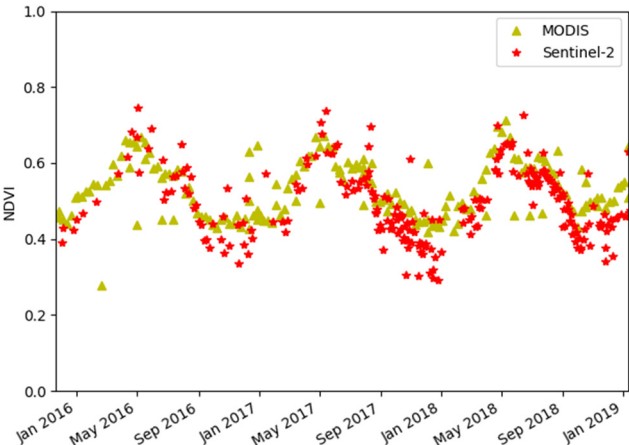

**Figure 3.** NDVI computed from MODIS and Sentinel-2 data over a 3-year period for the 31TCJ tile (near to Toulouse, in southwestern France).

It is then considered that for a given pixel, the NDVI can be expressed as the weighted sum of the NDVIs associated with each type of landscape, multiplied by the proportion of each of these landscapes, as shown in Equation (4). The latter proportions are derived from the landcover map, at the resolution provided by Sentinel-2. Since the values of NDVI provided by MODIS and Sentinel-2 are similar, the NDVI for each 500 m × 500 m pixel can be computed from the mean value of the MODIS pixel NDVIs. As explained above, it was preferred to use the NDVI computed from MODIS data rather than from Sentinel-2 data, due to the higher expected useful coverage of all study sites by MODIS. Equation (4) is thus computed at a scale of 500 m.

$$NDVI_{\text{av}} = NDVI_{\text{forest}} \times F_{\text{forest}} + NDVI_{\text{water}} \times F_{\text{water}} + NDVI_{\text{urban}} \times F_{\text{urban}} + NDVI_{\text{other}} \times F_{\text{other}} \quad (4)$$

where:

- $F_{\text{forest}}$, $F_{\text{urban}}$, $F_{\text{water}}$, and $F_{\text{other}}$ are the percentages of Sentinel-2 pixels tagged as forest (and, respectively, urban, water, or other). As an example, for the 31TCJ tile, 13.3% of this area is covered by forests, 9.2% is urban, and only 0.9% is covered by water. Thus, the remaining approximately $\frac{3}{4}$ of the 31TCJ tile is tagged as agricultural or natural landscapes;
- $NDVI_{\text{av}}$ is the average value computed over 4 MODIS pixels (since 1 MODIS pixel is around 250 m × 250 m);

- $NDVI_{\text{other}}$ is the NDVI corresponding to agricultural fields;
- $NDVI_{\text{forest}}$ is the NDVI associated with pixels tagged as forest. Using Sentinel-2 NDVI products, it is possible to observe (Figure 4) that $NDVI_{\text{forest}}$ generally follows an annual cycle. It is straightforward to determine the annual cycle, which can be approximated by a 2nd-order polynomial;
- $NDVI_{\text{urban}}$ is the NDVI computed from Sentinel-2 data and associated with pixels tagged as urban. The urban NDVI remained almost stable throughout the measurement period;
- $NDVI_{\text{water}}$ is the NDVI associated with pixels tagged as bodies of water. As a wide variety of water-covered surfaces can be encountered, it is known that the associated NDVI can range between a negative number and 0.1. However, as the proportion of the 31TCJ tile's surface area covered by water is less than 1%, it was not possible to establish reliable statistics for $NDVI_{\text{water}}$.

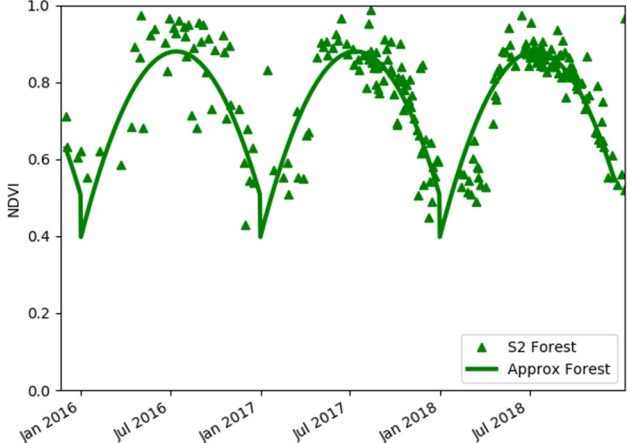

**Figure 4.** Variations of the $NDVI_{\text{forest}}$ measured by Sentinel-2 over a 3-year period for the 31TCJ tile.

On the basis of these results, it was decided to compute a corrected NDVI at a resolution of 500 m using Equation (5), which represents the average combined NDVI of the agricultural and natural landscapes. The NDVI contribution from water surfaces is neglected in this expression, since its contribution is very small.

$$NDVI_{other} \approx \frac{NDVI_{av} - NDVI_{forest} \times F_{forest} - NDVI_{urban} \times F_{urban}}{F_{other}} \tag{5}$$

Sentinel-2 data can thus be used to accurately characterize the forest NDVI cycle and the (virtually constant) NDVI in urban areas.

The use of distinct NDVI indexes to characterize different types of land cover, as described in Equation (5), was validated on French sites only, due to the strong heterogeneity of land cover in this part of the world.

In the case of our African sites, which are homogeneous, have no urban areas and are covered almost exclusively by agricultural areas or dispersed vegetation cover, an average value of NDVI is computed over 4 MODIS pixels, in order to determine the NDVI at regular 500 m intervals:

$$NDVI_{\text{other}} = NDVI_{\text{av}} \tag{6}$$

### 3.4. Use of Seasonal Surface Soil Moisture Information

In Equation (2), the minimum and maximum values of surface soil moisture ($\theta_{\min}$ and $\theta_{\max}$) are determined for the full period of ground measurements (3 years), and no seasonal variations are taken into account.

In the following we present the example of the Mouthoumet site. As can be seen in Figure 5, the ground data distribution has two distinct peaks, corresponding to the winter (wet) and summer (dry)

seasons, and is not just a simple Gaussian distribution. In order to provide a valid surface soil moisture estimation, extreme values should thus be computed for each of these two periods. These two seasons are indeed characterized by different rates of evaporation, which lead to different vertical surface soil moisture profiles. We propose to use the air temperature to distinguish between the winter (colder) and summer (warmer) seasons at this site, with the cutoff between these two seasons being defined by the median annual temperature. Statistics of air temperature conditions are considered to allow to discriminate the two seasons. They can be obtained from the World Meteorological Organization service at http://worldweather.wmo.int/en/home.html.

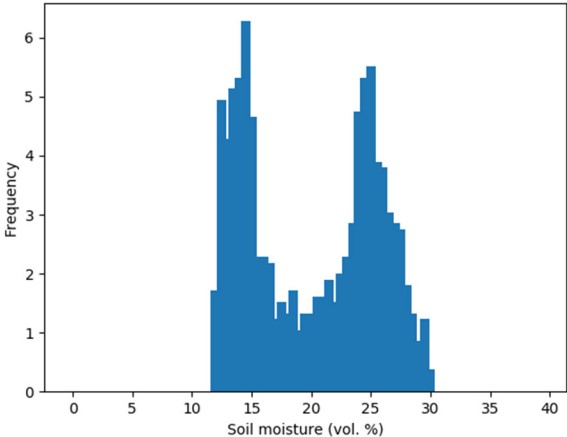

**Figure 5.** Distribution of surface soil moisture over the Mouthoumet (MTM) site (France) for the 2016–2018 period.

However, in Niger and Benin, rainfall occurs during the warmer monsoon season, whereas there are no rainfall events, and relatively small temporal variations in temperature during the winter season. For this reason, there is no physical sense in applying this procedure for the Niger and Benin sites, and just one set of minimum/maximum radar signals was computed. However, in the case of Tunisia and France, where two seasons are indeed observed, a pair of min/max surface soil moistures was chosen as a function of the day of the year.

The diagram in Figure 6 summarizes the processes applied by the SMES algorithm, as a function of studied area. Option 1 is the one applied to the Beninese and Nigerian sites; it corresponds to the general methodology. For the Tunisian sites, Option 2 is applied; that means that the radar signal is corrected, and that the seasonal surface soil moisture information is used. We can see that there are two min/max ground measurements (one red corresponding to the hot season and one blue to the cold season). Finally, Option 3 used for French sites adds to Option 2 the vegetation density information use. The light green on the curve of the NDVI represents the low NDVI levels and the dark green the high NDVI levels. Then, two min/max couples of the radar signals are computed and, as a function of the date, one or the other couple is chosen.

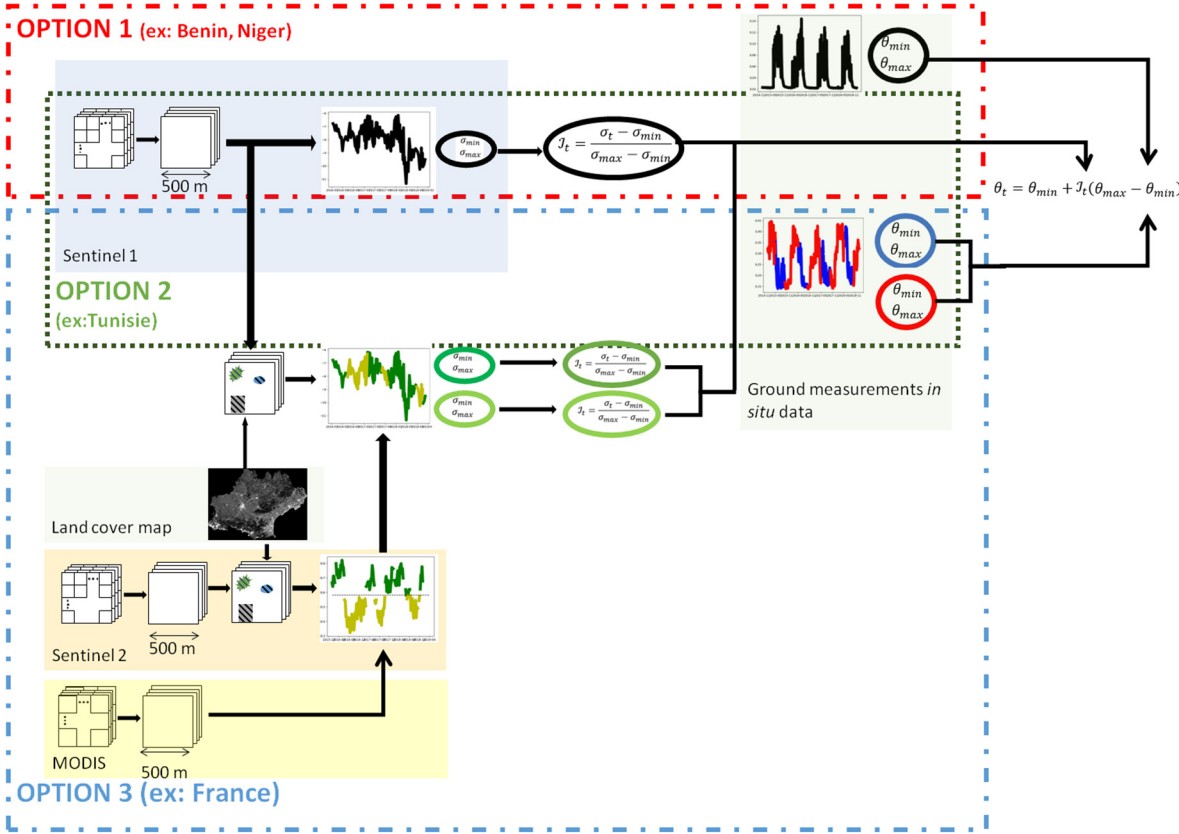

**Figure 6.** General scheme of the Soil Moisture Estimations from the Synergy of Sentinel-1 and optical sensors (SMES) algorithm.

## 4. Results

The improvements contributed by each of the optional processes (1, 2, and 3, described in detail in Section 3), which were added to the general detection algorithm, are analyzed in the following. The general algorithm was maintained for the Niger and Beninese sites, but in the case of the French and Tunisian sites, several processes were added to the general approach, in order to refine the results. For each process, the RMSE between the SMES estimations and the ground measurements were compared, with and without each of the relevant options.

### 4.1. Contributions of the Three Proposed Processes

The Savenès site serves to illustrate the application of the first of these optional processes ("radar signal correction"). This consists of computing an average radar value, which excludes contributions from forest, urban, and water-covered areas. If this process is not applied, all of the observed pixels are considered, with no distinction being made in terms of their type of land cover. In the case of the ascending orbit only, the computed RMSE is equal to 9.2 vol.% when this first option is applied, and 9.8 vol.% when it is not applied. The potential gain is thus 0.6 vol.%. However, the outcome resulting from the implementation of this process depends on the site and the time of year. The difference gain in performance is more pronounced in the months of March and October, which can be explained by rapid variations in vegetation properties at these times of the year. The second optional process ("use of the vegetation density information") is applied for all of the French sites. Then, a significant improvement in RMSE could be achieved by separating the radar signals into two vegetation classes, as a function of NDVI. As an example, the gain in RMSE was equal to 0.6 vol.% for the Narbonne site.

As described in Section 3.4, the third optional process ("use of seasonal surface soil moisture information") involves the use of two seasonally discriminated pairs of minimum/maximum values of

in situ surface soil moisture. For all French and Tunisian sites, it seems appropriate to consider two seasons. This means that the min/max of the in situ, cold winter season surface soil moisture is used for the remotely sensed surface soil moisture estimations during the winter, and a similar methodology is applied during the summer season. As an example, the gain in RMSE using this approach was equal to 0.7 vol.% for the Mouthoumet site.

The use of these processes for some of the study sites clearly allows the accuracy of the estimated surface soil moisture to be improved. In addition, it would be reasonable to assume that this gain in performance could be improved through the use of a longer time series of in situ surface soil moisture data. Indeed, all of these processes are based on the use of time series statistics: min/max of the radar signal, min/max of in situ measurements for Option 3, and two vegetation classes for Option 2.

## 4.2. Validation of Surface Soil Moisture Estimations Determined with the SMES Algorithm

Figures 7–10 compare the true surface soil moisture, given by the in situ measurements (green curve), with the surface soil moisture estimations produced by the SMES algorithm (ascending orbit shown by red stars and descending orbit by blue points). For each simulation, one polarization is chosen (for example VV) as well as one type of orbit (for example ascending). In addition, for each site, data from two Sentinel-1 ascending orbits (orb1 and orb2), corresponding to two different incidence angles, is used. Thus, at time $t$, if the Sentinel-1 image is observed from orbit orb1, $\mathcal{I}_t$ is computed using orb1 images only (such that no orb2 images are used), or inversely. This approach certainly reduces the number of images used by the change detection application, and prevents errors resulting from an empirical incidence angle normalization that could change from one pixel to another, and from one season to another. These results are compared with the ASCAT surface soil moisture product converted in volumetric moisture by using ground soil moisture time series, represented by black points in Figures 7–10. ASCAT soil moisture products as SMES algorithm products are retrieved for one pixel around ground measurements. The results shown in these figures were generated using data recorded in the VV polarization. In addition, the values of RMSE, R, bias, and ubRMSE obtained with the optional processes are listed in Table 3.

**Table 3.** RMSE, R, bias, and ubRMSE on surface soil moisture (vol.%) for all sites (VV polarization).

| | | | RMSE | | R | | Bias | | ubRMSE | |
|---|---|---|---|---|---|---|---|---|---|---|
| | | | SMES | ASCAT | SMES | ASCAT | SMES | ASCAT | SMES | ASCAT |
| Niger BZ1 | ASC | General | 1.6 | 1.6 | 0.62 | 0.70 | 0.0 | 0.4 | 1.6 | 1.6 |
| Benin BLM | ASC | General | 4.4 | 4.4 | 0.91 | 0.94 | −0.6 | −3.1 | 4.4 | 3.2 |
| Tunisia INGC | ASC+DES | General | 5.4 | 7.3 | 0.65 | 0.47 | −0.7 | 4.3 | 5.3 | 5.9 |
| | ASC | + Proc | 5.5 | | 0.62 | | 0.1 | | 5.5 | |
| | DES | 3 | 5.2 | | 0.69 | | −1.4 | | 5.0 | |
| Tunisia Hmidate | ASC+DES | General | 2.8 | 2.5 | 0.39 | 0.36 | −0.6 | 1.1 | 2.8 | 2.2 |
| | ASC | + Proc | 2.9 | | 0.26 | | −0.3 | | 2.9 | |
| | DES | 3 | 2.8 | | 0.51 | | −1.0 | | 2.6 | |
| France Narbonne | ASC+DES | General | 7.2 | 4.8 | 0.08 | 0.65 | −1.6 | 0.8 | 7.0 | 4.7 |
| | ASC | + Proc | 7.3 | | 0.07 | | −1.7 | | 7.1 | |
| | DES | 1/2/3 | 7.2 | | 0.08 | | −2.0 | | 6.9 | |
| France Savenes | ASC+DES | General | 9.9 | 9.7 | 0.23 | 0.39 | −2.6 | −2.4 | 9.6 | 9.4 |
| | ASC | + Proc | 9.2 | | 0.32 | | −2.1 | | 8.9 | |
| | DES | 1/2/3 | 10.7 | | 0.14 | | −2.1 | | 10.2 | |
| France Mouthoumet | ASC+DES | General | 3.4 | 4.2 | 0.78 | 0.64 | −0.3 | 0.4 | 3.4 | 4.2 |
| | ASC | + Proc | 3.0 | | 0.82 | | −0.0 | | 3.0 | |
| | DES | 1/2/3 | 3.9 | | 0.70 | | −0.9 | | 3.8 | |

### 4.2.1. Niger

The Niger site has a semi-arid climate (maximum in situ surface soil moisture measurement around 14 vol.% with sandy soil conditions). As can be seen in Figure 7, the results obtained at this site are very good (RMSE around 1.6 vol.%, ubRMSE equal to 1.6 vol.%, and bias equal to 0). The correlation coefficient R between SMES product and ground measurement is moderate, equal to 0.62.

The surface soil moistures computed with the SMES algorithm are very similar to those provided by the ASCAT product. During the dry season, the soil moisture is very low. Monsoon season presents large temporal variations of soil moisture, which are fairly well retrieved by satellite estimates. The texture of the soil is sandy so that the maximum soil moisture is relatively low, and the soil moisture can vary very quickly. This can make the soil moisture estimate at a 12-day time resolution relatively not sufficient for hydrological applications.

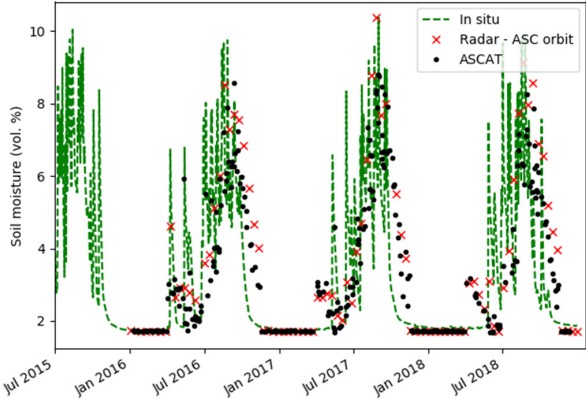

**Figure 7.** Comparison of surface soil moisture from SMES and ASCAT products, ground measurements over the BZ1 site (Niger).

### 4.2.2. Benin

Similarly to the Nigerian site, the Beninese site is found to have good results (Figure 8) with an RMSE of approximately 4.4 vol.%, an ubRMSE equal to 4.4 vol.%, a high correlation coefficient between estimations and ground measurements (R equal to 0.91). When the SMES and ASCAT products are compared, the latter surface soil moisture estimation can be seen to be slightly better. This could be explained by the short duration of the Sentinel-1 time series. The precipitation level is higher in this site than in the Niger one, and thus the wet monsoon season is marked almost continuously with high moisture values. Satellite estimates reproduce this cyclical aspect of soil moisture very well.

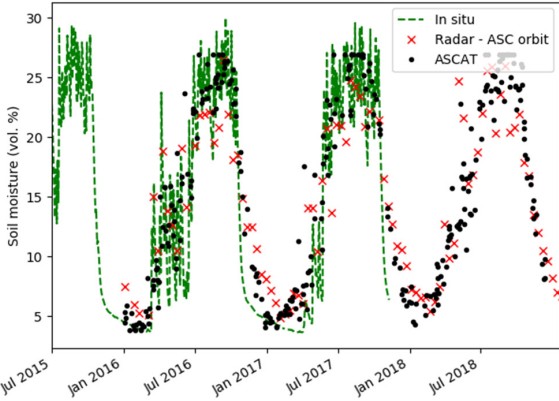

**Figure 8.** Comparison of surface soil moisture from the SMES and Advanced Scatterometer (ASCAT) products, of ground measurements over the BLM site (Benin).

### 4.2.3. Tunisia

At the INGC site (Figure 9), the highest RMSE (5.5 vol.%) is observed in the case of the ascending orbit. A moderate correlation coefficient approximately equal to 0.65 is observed between the SMES and ground measurements. Furthermore, the presence of two seasons plays a key role at this site, since the strong changes in evaporation between the cold and hot seasons are taken into account through the use of the 3rd optional process. This context is characterized by a high evaporation level and dispersed precipitation events, which produces large temporal variations in soil moisture throughout the year. This can reduce the accuracy of the daily satellite estimates.

At the Hmidate site, the same very low value of RSME (approximately equal to 2.9 vol.%) is obtained for the ASCAT and SMES algorithms. The correlation coefficient (R = 0.39) between SMES products and ground measurements is low. This can be explained by a high temporal variations of soil moisture in a context of sandy soils and high evaporation levels.

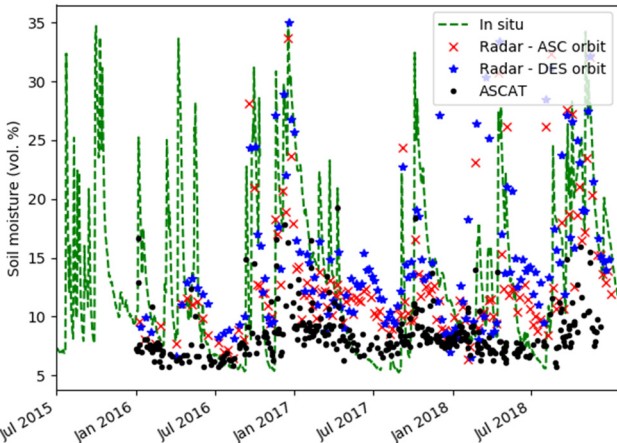

**Figure 9.** Comparison of surface soil moisture from SMES and ASCAT products, of ground measurements over the INGC site (Tunisia).

### 4.2.4. The Occitania Region (South of France)

At the sites in the south of France, the results are less accurate than those obtained for the African sites. This can be explained by the strong heterogeneities of land cover, with less frequent flat landscapes. The RMSE can still be very satisfactory to satisfactory.

As shown in Figure 10, under relatively hot climatic conditions (summer), the surface soil moisture estimations are well matched with the true surface soil moisture. At other times of the year, the SMES surface soil moisture estimations are still robust, although considerably less accurate. This induces a low correlation coefficient (R lower than 0.32) for two studied sites. This could be explained by strong, rapid temporal variations in vegetation that can affect the implementation of the change detection technique, as well as by certain agricultural practices, such as vines being strung along metal wires, which are frequently encountered on these sites. The moderate accuracy of these results was also retrieved in [54].

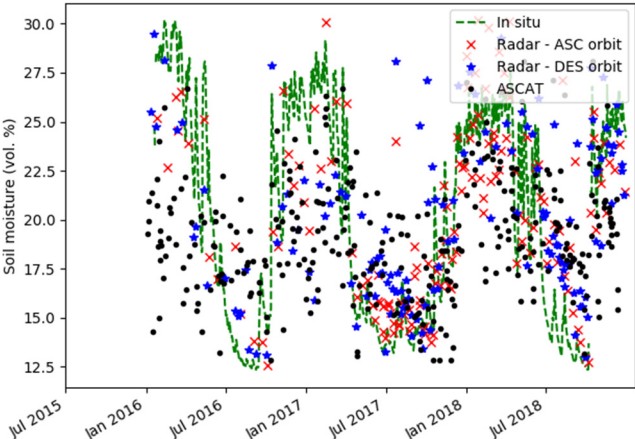

**Figure 10.** Comparison of surface soil moisture from SMES and ASCAT products, ground measurements, over the MTM site (France).

For all sites, we observe approximately the same levels for RMSE and ubRMS. This is due to the fact that proposed SMES volumetric moisture values are retrieved using a calibration with ground measurements. Therefore, the bias is generally close to zero.

At all of the study sites, the ground measurements were made at a depth close to 5 cm, whereas the radar surface soil moisture estimations correspond to a depth of approximately 2 cm, or even 1 cm under wet conditions. This discrepancy could explain some of the differences observed between the in situ measurements and the SMES surface soil moisture estimations.

The influence of soil roughness is not taken into account with this approach. This is a priori a valid assumption for the agricultural sites that were tested. On sites characterized by very large plots, the soil roughness could be expected to have a small influence on the backscattered radar signals.

It is interesting to note that the moisture index, which ranges between 0 and 1, is estimated directly from satellite data, with no need for local field measurements. With the exception of Option 3, the soil porosity characteristics could be used to identify the wilting and field capacity points when converting the moisture index to volumetric moisture, for which the minimum and maximum values of soil moisture are needed. The latter approach would make it possible to generalize the use this relatively simple algorithm. Although it would allow more accurate estimations of the volumetric moisture to be made, the implementation of this conversion in the case of Option 3, with a seasonal separation, would be dependent on ground measurements of field moisture. This requirement certainly represents a shortcoming of the proposed methodology in cases where it is applied to sites on which there are no field measurements.

Comparable results are obtained when the same analyses are carried out using VH polarization data.

## 5. Conclusions

This paper presents the SMES algorithm, which can be used to estimate surface soil moisture through a synergistic combination of Sentinel-1 radar data and optical measurements. The novelty of this approach resides in the fact that it estimates surface soil moisture at a spatial resolution of 500 m, and a temporal resolution of 6 days, using a change detection approach based on land cover heterogeneity. A good knowledge of the topography, climate, and vegetation of the studied regions is essential when it comes to understanding variations in their surface soil moistures. In addition to the general change detection methodology, three optional processes are proposed. The first of them is used to compute the radar signal only over those areas having no urban or forest cover. The second process separates the radar image pixels into two classes, depending on the corresponding vegetation characteristics (based on NDVI level), before applying the change detection algorithm. The third

process considers two climatic seasons, associating each of these with minimum and maximum values of in situ surface soil moisture, which are used to convert the computed moisture index to volumetric moisture. According to the context of each studied site, the SMES algorithm can then be adapted by applying none, one, or several of these processes.

To validate this methodology, its surface soil moisture estimations were compared with in situ surface soil moisture measurements at several test sites (Occitania region in France, Merguellil region in Tunisia, Dallo Bosso in Niger, and Ouémé in Benin), characterized by different soil and climatic conditions, and over a period of more than 3 years. The results obtained by applying the general change detection algorithm are satisfactory (RMSE lower than 2 vol.% for Niger and lower than 5 vol.% for Benin). For the Tunisian sites, when the 3rd additional process is used to improve the general algorithm, the RMSE between in situ measurements and SMES estimations is low (5.2 vol.% for the INGC site and less than 2.9 vol.% for the Hmidate site). At the French sites, due to the presence of more complex landscapes, the RMSE between ground measurements and SMES estimations ranges between 3.4 vol.% and 9.9 vol.%. Overall, the ASCAT and SMES products are comparable in terms of their performance in the estimation of surface soil moisture. As the SMES algorithm is based on a change detection technique, it is reasonable to expect that its performance could improve after a period of several years, whereas the results presented in this study were derived, in the best case, from less than 4 years of Sentinel-1 data. This approach also could be generalized by applying Cumulative Density Function (CDF) matching, which is more sophisticated than a simple hypothesis of linearity between soil moisture and the radar signal [61]. High resolution approaches also should be more generalized in a context of increasing processing capabilities.

**Author Contributions:** M.Z. and M.F. designed the methodologies; M.F. developed the algorithms; N.B. provided advices on results interpretation; C.A., T.P. and J.-C.C. processed the ground measurements; M.F. and M.Z. wrote the paper; all authors reviewed the paper. All authors have read and agreed to the published version of the manuscript.

**Funding:** This research was funded by the RTRA POMME-V, CNES/TOSCA TAPAS, and 4000129870/20/I-NB ESA programs.

**Conflicts of Interest:** The authors declare no conflict of interest.

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
