# Peer review of "Estimating 500-m Resolution Soil Moisture Using Sentinel-1 and Optical Data Synergy"

_water, doi:10.3390/w12030866_

Round 1
Reviewer 1 Report
The authors have used S1, S2 and MODIS data to produce 6-days soil moisture with 500m spatial resolution. I still do not understand what the advantages of this combination is, because the temporal resolution is 6 days (the same as S1 and S2), while the spatial resolution reduced from 10 m to 500m! What is the logic and advantage of this combination?
The novelty of this study is nor clearly mentioned. It seems the authors have used a general remote sensing approach to estimate SM.
More literature review and discussion are required in the Introduction section
Figure 1 is not clear
More discussion is required in section 4. It’s now mostly reporting the results.
Author Response
Answer in file

Reviewer 2 Report
Estimating soil moisture from active radar measurements under highly heterogeneous regions is still challenging. The paper titled “Estimating 500m resolution soil moisture using Sentinel-1 and optical data synergy” developed a method to estimate surface soil moisture based on the conventional change detection technique by combining use of Sentinel-1 and optical (Sentinel-2 and MODIS) data. Particularly, the authors proposed three steps to deal with the influence of land cover heterogeneity, vegetation and seasonal variation of soil moisture to further improve the accuracy of soil moisture retrievals. The results were finally validated by in situ measurement from several sites in France and Africa and also compared with the ASCAT soil moisture products. The following issues should be carefully addressed before publication in Water.
General Comments:
1) The most serious thing seems the possibility of applying the method at a global scale or other regions where in situ measurements are not available. The proposed method relies heavily on the in situ measurements. First, they need in situ measurements to convert the soil moisture index (0~1) retrieved from the change detection method to volumetric soil moisture with physical unit of m3 m-3. Second, to perform option 3, prior knowledge (e.g., the distribution of ground measurements) is needed to determine the seasonality of soil moisture. Besides, from table 3, it is seen that different schemes were adopted in different sites. How do others know which scheme (option) should be adopted for their interest area if the ground measurements are not available? When should one use option 1/2/3 or their combinations? These issues should be fully discussed.
2) It is not clear what are the advantage and disadvantage of your proposed index compared with the ASCAT-based soil moisture index since they all used the change detection method. Previous studies show that ASCAT soil moisture performs better than passive radiometer-based soil moisture products in densely vegetate areas. Do the authors know the reason since you also proposed option 2 to deal with the effects of vegetation?
3) The authors used the maximum and minimum soil moisture values to convert soil moisture index to volumetric soil moisture (equation 2). Did you test other more commonly used scaling techniques such as the CDF matching approach which may get more accurate results?In other word, did you think the Min/Max method is superior to the CDF matching method?
Specific Comments:
1) Line 26: Please add a unit for RMSE.
2) Line 63: Please give the full name of DISPATCH.
3) Line 84: Please provide the data portal for all the data (e.g., in situ measurements, MODIS, ASCAT, etc.) used in this study.
4) Figure 1: Please improve the visibility (or DPI) of Figure 1 to make it clear.
5) Line 96: Please rephrase this sentence.
6) Line 107-108: Why did you choose these three stations?
7) Line 191 to 199: Did you use the ASCAT swath data or gridded data? How did you find the ASCAT grid to match the Sentinel grid with 500 m resolution?
8) Table 1: Please add the information of ASCAT.
9) Line 218: How did you aggregate Sentinel-1 pixels to a spatial resolution of 500 m?
10) Line 226: How did the time period used for determination of the minimum and maximum radar signals affect the final results? If one year or two years data are selected, did the results change much?
11) Figure 2: please add a legend.
12) Line 268: Please rephrase this sentence.
13) Line 274 to 286: Why the influence of surface roughness is not considered?
14) Line 305: Did you do some pre-processing (e.g., smoothing and interpolation) for the MODIS and Sentinel-2 NDVI data? How did you deal with the influence of cloud?
15) Line 355: did you use the 2nd order polynomial to fit the variations of the NDVI(forest) in your method?
16) Line 386: Where does the air temperature come from? Please clarify.
17) Figure 5: What is the corresponding distribution of radar backscatter?
18) Figure 6: It seems that the same maximum and minimum backscatter are used to calculate the soil moisture index (ℐt) when adopting option 2, while different maximum and minimum backscatter are used when adopting option 3. Why?
19) Line 413: It is not clear why the authors only adopted the single RMSE as their error metric. Other metrics such as ubRMSE, Bias and R should be provided. Since you only use measurements from a single station to validate the satellite soil moisture retrievals (particularly the ASCAT SM), the ubRMSE is more suitable than RMSE to quantify their accuracy.
20) Line 485: Please rephrase this sentence.
Author Response
Answer in file

Round 2
Reviewer 1 Report
The authors have addressed most of my comments. However, I still think the manuscript can be improved by providing more explanation in the Introduction section by (1) explaining why reducing 10m spatial resolution to 500m resolution can be beneficial and (2) capability of some big data processing platforms like GEE to produce the high spatial resolution soil moisture products.
Also, I think still the authors can provide more discussion in the result section on the limitation/advantages of their method.
Finally, I should mention that the manuscript contains over 20 self-citations (e.g., 1/3 of the references are from Ziribi)! The authors may be experts in this field, but they are better to pay more attention to the works of others.
Author Response
Reviewer1
We would like to express our sincere gratitude for the effort and all constructive comments made in reviewing this manuscript. We have tried to consider all of the proposed comments to make a revised version to meet the publication requirements.
- The authors have addressed most of my comments. However, I still think the manuscript can be improved by providing more explanation in the Introduction section by (1) explaining why reducing 10m spatial resolution to 500m resolution can be beneficial and (2) capability of some big data processing platforms like GEE to produce the high spatial resolution soil moisture products.
Answer : We thank reviewer for this remark, we add more details about this point.
In introduction :
“On the other hand, an intermediate resolution close to 1 km appears to be suitable for the analysis of hydrological processes in a regional context. This scale has thus been addressed in various studies [51-54]. This intermediate spatial scale can be used to bridge the gap between high resolution and low resolution products. The first ones need important information about land use, vegetation properties at field scale to retrieve soil effect on radar signal with important capacities of computing for large regions. Low resolution offers very useful information at very large scales (continental, global…), but is not accurate enough to analyze hydrological processes at regional scale. “
In conclusions:
“This approach could be also generalized by applying Cumulative Density Function (CDF) matching, more sophisticated than a simple hypothesis of linearity between soil moisture and the radar signal [61]. High resolution approaches should be also more generalized in a context of increasing processing capabilities.”
- As asked by reviewer, we propose more details in result section for different site.
- Finally, I should mention that the manuscript contains over 20 self-citations (e.g., 1/3 of the references are from Ziribi)! The authors may be experts in this field, but they are better to pay more attention to the works of others.
Answer : We agree with reviewer, we choose spontanously our publications as reference. We reduce this high percent in the new version, we consider other references coming from other teams.
Reviewer 2 Report
The authors have addressed most of my previous concerns, and the following minor comments should be further explained and some errors should be corrected before publication.
1) In the response, the authors stated that “For ASCAT, we consider also, the gridded data with the pixel covering the considered ground measurements”, while in the manuscript, it is mentioned that “ASCAT soil moisture products, recorded over a 25 km swath grid, can be recovered from the Earth Observation Portal (https://eoportal.eumetsat.int/)”. Did the authors convert the ASCAT swath data to the gridded data? Please give more details how did you attain this. Moreover, explain how did you find the satellite grid to match the corresponding in situ site (e.g., did you use the minimum distance between the satellite pixel and in situ site to determine the specific satellite grid for validation?).
2) The original ASCAT data downloaded from the Earth Observation Portal (https://eoportal.eumetsat.int/) is the soil moisture index range from 0 to 1. How did you convert the ASCAT soil moisture index (0~1) to the volumetric soil moisture with physical unit of m3 m-3? If you have used the soil porosity data, please give the source of the porosity data.
3) The data portal of the in situ measurements is still not given, please add this information.
4) Table 1: Please check the temporal resolution of ASCAT since it observes the earth surface with a very coarse resolution of ~ 25 km.
5) Line 420-421: Please clarify the source (e.g., from the sites) of the air temperature used in the study.
6) In the response, the authors stated that in the case of Figure 5, the corresponding distribution of radar backscatter (in dB scale) is Gaussian. However, it is clearly seen in Figure 5 that the distribution of soil moisture is not Gaussian. That is why the authors consider the seasonality of soil moisture in their algorithm. Did you confirm the distribution of radar backscatter in this case is Gaussian?
7) Table 3: it seems there are some errors in the error metrics. For example, in Niger BZ1, ubRMSE of ASCAT (1.6%) is even larger than the RMSE value (1.5%) which is not correct since the ubRMSE must be smaller than or equal (if the bias is zero) to RMSE. In Tunisia INGC, the error metrics seems also not correct. Please recheck all the results carefully.
8) Please add some description of other metrics (e.g., ubRMSE, R, Bias) rather than only RMSE in the results part.
Author Response
Reviewer2
We would like to express our sincere gratitude for the effort and all constructive comments made in reviewing this manuscript. We have tried to consider all of the proposed comments to make a revised version to meet the publication requirements.
The authors have addressed most of my previous concerns, and the following minor comments should be further explained and some errors should be corrected before publication.
- In the response, the authors stated that “For ASCAT, we consider also, the gridded data with the pixel covering the considered ground measurements”, while in the manuscript, it is mentioned that “ASCAT soil moisture products, recorded over a 25 km swath grid, can be recovered from the Earth Observation Portal (https://eoportal.eumetsat.int/)”. Did the authors convert the ASCAT swath data to the gridded data? Please give more details how did you attain this. Moreover, explain how did you find the satellite grid to match the corresponding in situ site (e.g., did you use the minimum distance between the satellite pixel and in situ site to determine the specific satellite grid for validation?).
Answer : The ASCAT soil moisture product "ASCAT Soil Moisture at 25 km Swath Grid - Metop" is not based on a grid as Sentinel or MODIS. So in our work, for each site and for each day with available data, an average value is computed over ASCAT measures in a region of +/-0.05° in lattitude and longitude around the site.
2) The original ASCAT data downloaded from the Earth Observation Portal (https://eoportal.eumetsat.int/) is the soil moisture index range from 0 to 1. How did you convert the ASCAT soil moisture index (0~1) to the volumetric soil moisture with physical unit of m3 m-3? If you have used the soil porosity data, please give the source of the porosity data.
Answer : For our case, we don’t consider porosity properties because we have ground moisture measurements during all the studied period. This aspect is clarified in the text.
« These results are compared with the ASCAT surface soil moisture product converted in volumetric moisture by using ground soil moisture time series, …. »
3) The data portal of the in situ measurements is still not given, please add this information.
Answer : We add this information for the three sites.
4) Table 1: Please check the temporal resolution of ASCAT since it observes the earth surface with a very coarse resolution of ~ 25 km.
Answer : There was an error in the first version. The temporal resolution of proposed data is approximately 3 days.
5) Line 420-421: Please clarify the source (e.g., from the sites) of the air temperature used in the study.
Answer : we add this information :
«Statistics of air temperature conditions are considered to allow to discriminate the two seasons. They can be obtained from the World Meteorological Organization service in http://worldweather.wmo.int/en/home.html.».
6) In the response, the authors stated that in the case of Figure 5, the corresponding distribution of radar backscatter (in dB scale) is Gaussian. However, it is clearly seen in Figure 5 that the distribution of soil moisture is not Gaussian. That is why the authors consider the seasonality of soil moisture in their algorithm. Did you confirm the distribution of radar backscatter in this case is Gaussian?
Answer : We are sorry, we thought that question made in first review is general to a distribution of radar signal for an homogenous area. No, distribution of radar signal during all the period is not gaussian. It illustrates variation of radar signal function of soil moisture. This is an illustration of one radar distribution.
7) Table 3: it seems there are some errors in the error metrics. For example, in Niger BZ1, ubRMSE of ASCAT (1.6%) is even larger than the RMSE value (1.5%) which is not correct since the ubRMSE must be smaller than or equal (if the bias is zero) to RMSE. In Tunisia INGC, the error metrics seems also not correct. Please recheck all the results carefully.
Answer : We thank reviewer for this remark. We off course agree with this remark. Some changes in data are considered in calculation of ubRMSE in the first version. Correction is made for all cases. For all sites, we observe approximately the same levels for RMSE and ubRMS. This is due to the fact that proposed SMES volumetric moisture values are retrieved using a calibration with ground measurements. Therefore, the bias is generally close to zero.
8) Please add some description of other metrics (e.g., ubRMSE, R, Bias) rather than only RMSE in the results part.
Answer : As asked by reviewer, we add description of the different metrics in results section.